# Were Culture and Heritage Important for the Resilience of Tourism in the COVID-19 Pandemic?

**Krešimir Jurlin**

Research Associate, Institute for Development and International Relations, Lj. F. Vukotinovića 2, 10000 Zagreb, Croatia; kreso@irmo.hr; Tel.: +385-1-487-7479

**Abstract:** The unprecedented impact of the COVID-19 on the world tourism is clear and obvious. Still, modelling the impact on individual countries faces many problems from data availability to the multitude of underlying variables rather difficult to capture. This study used simple and multiple regression to research possible effects of the recent pandemic to the fall in the volume of tourism in 20 European countries, throughout the 20-month period. The results of this study were rather surprising showing that the relative fall in tourism cannot be explained only by incidence of COVID-19 by countries, while in multiple regression by adding the variables of distance of travel and composition of tourism by facilities coefficients of determination were very low. Adding variables of natural and cultural heritage as well as of cultural activities somewhat improved the baseline model with the best fitting variable of culture visits adding 11.8 percentage points to the explanatory power of the model, while culture employment and culture consumption added a possibly important 5.6 and 2.6 points, respectively. Although these findings are in line with recent literature of resilience and changes in tourism due to pandemic, a more thorough research is needed to further investigate these relations.

**Keywords:** impact of COVID-19; travel; wellbeing; NATURA; UNESCO

## 1. Introduction

The fall of tourism due to recent pandemic was overwhelming and unprecedented, which especially stands for international arrivals falling by 73% in 2020, virtually remaining the same in 2021, and not likely to return to pre-crisis levels until 2024 (UNWTO 2022). Total volume of tourism decreased somewhat less than arrivals, especially in countries that could have compensated inbound tourism by domestic tourists staying within national borders (UNWTO 2022). With a widespread vaccination as well as some adjustments of destination choices and facilities, providing less risk of close contacts with other persons, the volume of European tourism remained at the level considerably higher than the world average. With a fall in the numbers of COVID-19 cases during the summer of 2021, volume of domestic tourism reached the levels even higher than in the same period of 2019, while the inbound tourism followed at a lower level between 60% and 70%, compared to 2019 (Appendix A Figure A1).

While inbound tourism (number of nights spent at tourist accommodation establishments) in the European Union (EU) countries fell considerably in 2020, to 31.2%, only to recover to 41.5% in 2021, figures for domestic tourism were much better, i.e., 68.1% and 80.9%, respectively. Therefore, decline in tourism in 2021 in the European countries was lower than the global decline, with the total volume recovered from 50.5% in 2020 to 62.1% in 2021, as compared to 2019 (Appendix A Table A1).

Among the tourist facilities, the least impact was on the nights spent at camping grounds, recreational vehicle parks and trailer parks with a fall to 70.3% to recover to 87.6% in the 2 years under review, followed by holiday and other short-stay accommodations with 57.3% and 69.2%, respectively. In contrast, nights spent in hotels and similar accommodation fell to a low level of 44.1% in 2020 to recover to 54.3% in 2021, indicating



that countries with very high share of hotels in the overall capacity of the accommodation establishment could have faced more significant impact. Most likely, hotels were perceived by tourists as bearing more risk of close contacts with other persons than while camping or in rented apartments and other short-stay accommodations.

According to the European Travel Commission European Travel Commission Report (2022) the outlook for 2022 is positive with tourism indicators reaching 80% of the pre-pandemic levels in 2022, and domestic tourism even exceeding the 2019 levels as a consequence of substitution by the intraregional tourism of the long-haul travel while tourists still keep the cautious and safe approach. Therefore, Europe shall show the best 2022 figures among the main tourist regions of the world. However, there are indications that the shock from the COVID-19 pandemic could remain permanent, and the model of tourism therefore could transform to a more sustainable one (Payne et al. 2021).

This study is focused on identifying possible impacts of culture and heritage on tourism outcome by countries, trying firstly to find out to what extent relative tourism performance may be attributed to relative intensity of COVID-19 by countries, followed by the analysis of the selected indicators of heritage and culture aiming to e find out their possible relationship to the relative fall in tourism by countries. The objective of the research is to contribute to the scientific literature of resilience of tourism as economic activity when faced with major crises. The paper is structured as follows: Section 2 provides an overview of the recent literature on impacts of the pandemic on tourism; Section 3 follows with the description of materials and methods used while Section 4 presents the main results of the analysis; Section 5 brings discussion of the findings towards the results obtained in similar studies, followed by the concluding remarks in Section 6.

## 2. Literature Background

In recent history, tourism was hit by several health effects (SARS, H1N1), cases of natural disaster, such as earthquakes, tsunamis and volcano eruptions, as well as wars, conflicts and political and security issues. However, risk management and long-term planning could reduce dependence on these external factors. Main impacts of these events on tourism were analyzed in scientific literature (Ma et al. 2020; Novelli et al. 2018; Rosselló et al. 2020) leading to conclusion that both natural disasters and crises caused by humans resulted in decline of tourist arrivals due to objective causes, i.e., formal obstacles to tourism industry activities, as well as to psychological risk perception by the tourists. Apart from that, there is also the "neighbourhood" effect (Maphanga and Henama 2019) whereby countries that are less or not at all affected, but are geographically close to those that are, also face reduced tourist arrivals.

COVID-19 pandemic shed new light on resilience to unpredicted and imposed challenges in the scientific literature. Some studies (Buzinde 2020; Wen et al. 2020; Garcês et al. 2020) focused on the importance of wellness and wellbeing as being more important for tourism within and after the world health crisis. Most likely, privacy, i.e., less contact with other tourists, as well as possibilities of spending time actively and in the natural surroundings may become more and more important (Santos et al. 2020). A well-known effect of the outdoor physical activities on psychological health seemingly also proved to contribute to mitigation of the negative effects of pandemic (Buckley and Westway 2020). This goes in line with research findings that for the future success of tourism, services should be tailored made, i.e., personalized (Abbas et al. 2021). Similarly, there is a noticeable preference given to private means of transport than using public transportation (Zheng et al. 2021). Moreover, many tourists postponed their planned trips even after the establishment of safe conditions. The regional and global crises have rather different effects across countries depending on health system performance, the intensity of the direct impact, as well as psychological uncertainty induced by the pandemic event (Aronica et al. 2021). The literature dealing with possible impacts of natural and cultural heritage on the resilience to crises is very limited and will be pointed out to in the discussion section.

## 3. Materials and Methods

Reliable and methodologically sound data sources were of the utmost importance for the methodology used (simple and multiple regression). The problem lies in different methods of gathering data on the intensity of pandemic by countries. There is a significant "grey area" of tourism statistics that concerns reporting on private accommodation and small-scale facilities as well as on houses and apartments owned by tourists themselves. Aside that, in many cases data collection methodology on the categories of overnight stays by the type of facilities differs considerably. Variables of type of transport used in tourism travel are not methodologically unified either, even for the EU countries. At the outset, the sample was meant to include European countries that report to Eurostat as much as possible. However, due to limited data availability, only 20 countries remained in the sample: Austria, Belgium, Croatia, Czechia, Denmark, Estonia, Finland, Germany, Hungary, Iceland, Italy, Latvia, Liechtenstein, Lithuania, Luxembourg, Malta, Netherlands, Norway, Poland, Portugal, Romania, Slovakia, Slovenia, Spain and Sweden. Dependent variable was the tourist overnight stays, calculated as 20 monthly indices during the pandemic (March 2020–October 2021) compared to the 2019 monthly data.

Firstly, simple regression of these data was conducted to show a direct impact of the monthly average data of new COVID-19 cases by country. After that, multiple regression was conducted by adding the indicators of the share of air transport in total inbound tourist arrivals as well as data on the share of hotels and similar accommodation in the total of monthly tourist overnight stays by countries. Finally, apart from the mentioned 3 variables of that baseline regression, further 5 multiple regressions were conducted, testing a possible additional explanatory power of natural and cultural endowments as well as cultural activity by countries to the variance of the said dependent variable representing outcome of tourism by countries. The methods were very basic, without a focus on t-stat and *p*-values. Moreover, independence of the variables used in multiple regression was only assumed, i.e., no multicollinearity check was conducted. All the details on the data used for the analysis (definitions and sources) are presented in the Data Availability Statement.

## 4. Results

The results of simple regression on impact of relative intensity of COVID-19 on tourist overnight stays for the 20 countries under review were rather surprising. The coefficients of determination for almost the entire 20-month period were extremely low. This indicates that variance in the relative fall in tourism cannot at all be explained only by different relative incidence of pandemic by countries. Multiple regression with two more variables assessing "closeness" of travel (share of inbound tourism conducted by land transport in total tourist arrivals) and composition of tourism facilities according to a perceived risk due to close encounters with other persons (share of hotels in total number of overnight stays) performed somewhat better in explaining relative fall in tourism by countries. Still, out of total of 20 months under review, only for 7 months there was at least a very small relation between the three variables used and the relative fall in tourism by countries. As evident from the Figure 1, the value of the coefficient was 0.1 or higher in the three summer months of the first year of the pandemic. This may be explained as in this period there was a significant fall in the pandemic intensity, while the tourists were somewhat cautious in their destination choices preferring close destinations as well as choosing type of accommodation that facilitates avoidance of close contact with strangers.

The second period with at least 0.10 coefficient of determination in multiple regression were the last 2 months of 2020 throughout the outburst of pandemic. The data for December may be attributed to the seasonal increase in travel for holidays with the large share of private travel, being somewhat more dependent on the risk factors than business travel. Surprisingly, throughout 2021, there was almost no relation of tourist overnight stays with relative intensity of pandemic, even when combined by the two additional variables in multiple regression. A possible reason may be that with a significant fall of pandemic intensity as well as with widespread vaccination, tourists largely returned to their previous

habits, although still at a lower level than before. A final period for which there is some statistical link between the variables employed to try to explain differences in tourism fall (now recovery) by countries are the last two months under review i.e., September and October of 2021. With a new increase of pandemic figures and regardless of a high percentage of vaccinated population, there was still a relatively strong impact of COVID-19 cases in relative tourism performance. However, it shall be made clear that even for the 7 months with detected impact on tourism, the coefficients are rather very low.

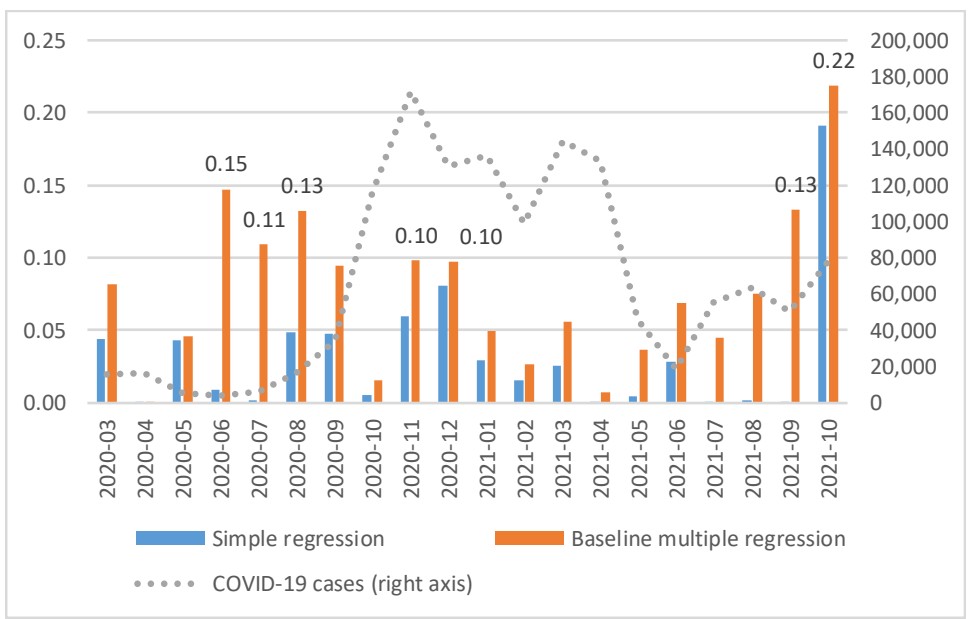

**Figure 1.** Coefficients of determination in simple regression and in baseline multiple regression and COVID-19 cases in 20 months for the 20 countries under review. Source: Authors' calculation, using data specified in the Data Availability Statement.

For further analysis of a possible effect of the selected indicators of culture and heritage out of the 20 periods under review only 7 were selected, with a baseline scenario multiple regression coefficients of determination 0.10 or above i.e., only for months with at least small statistical impact of pandemic on relative tourism performance by countries. The findings seem to be rather interesting indicating there was some impact of heritage and culture on relative fall of tourism. With only few exemptions, adding one more variable to three variables in the baseline scenario improved to a certain extent the explanatory power of the model, i.e., increasing the coefficient of determination (Figure 2).

Most notable impacts are for culture visits in 2020-6, 2020-11 and 2020-12, culture employment in 2020-6 and 2021-10 as well as for the UNESCO indicator for 2020-12. On average, the most notable impact on further explanation of tourism performance by countries was of the indicator of culture visits, while the lowest impact was of the indicator of culture consumption.

The main findings of the research are briefly presented in Figure 3. While the baseline multiple regression (COVID-19 cases, share of land travel and share of hotels) helped explaining additional 7.8% of the variance of the tourist overnight stays by countries as compared to a simple regression using only COVID-19 cases as independent variable, adding variable of culture employment as well as variable of the relative number of UNESCO protected sites added a possibly important 5.6 percentage points to the statistically explained share of the dependent variable. The strongest impact was of the variable of culture visits with 11.8 percentage points added to the explanatory power of the model, which, combined with the two variables of the baseline scenario, enabled the model to attribute a moderate 25.2% of the variance of the monthly tourism outcome by countries.

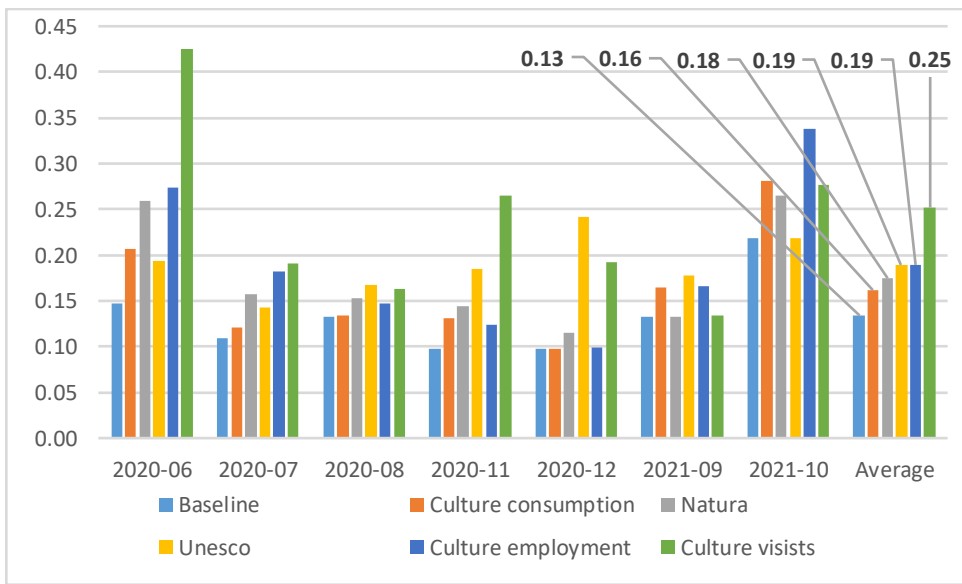

**Figure 2.** Coefficients of determination ($R^2$) for 7 months with at least 0.10 values in the baseline multiple regression, and further 5 regressions adding indicators of heritage and culture for the 20 countries under review. Source: Author's calculation, using data specified in the Data Availability Statement.

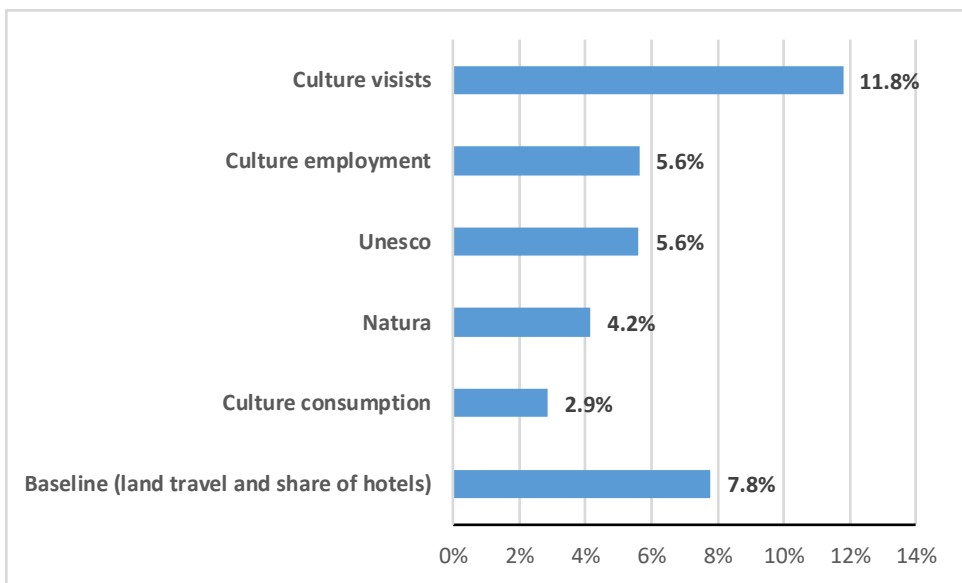

**Figure 3.** Percentage points of coefficients of determination added on average to the simple regression for 7 months by the baseline multiple regression, and further 5 regressions for the 20 countries under review. Source: Author's calculation, using data specified in the Data Availability Statement.

However, it should be noted that, while having a certain explanatory power, the two indicators of heritage endowment, i.e., percentage of terrestrial protected area ("Natura") and per capita number of World Heritage sites inscribed by each State Party ("Unesco"), have negative values of the respective coefficients in multiple regression so they cannot be considered as factors mitigating the impact of pandemic on tourism (Table 1).

Still the other indicators, i.e., visits to cultural sites (regardless of one case of negative value of the regression coefficient), culture consumption and employment might have, within the limits of the methodological approach of this study, some explanatory power showing that the relative intensity of cultural activities may have contributed to the better outcome, i.e., higher indices of inbound tourism. Again, it came as a surprise that each one of these indicators added more to the explanation of the variance of the tourism outcome

than the variables in the baseline regression using three variables presumably strongly connected to the depending variable, i.e., the incidence of the disease, the arrivals by land and the share of hotels in the pre-pandemic overnight stays.

**Table 1.** Coefficients of determination for 7 months with at least 0.10 values in the baseline multiple regression, and further 5 regressions adding indicators of heritage and culture for the 20 countries under review.

|  | Baseline | Natura | Unesco | Culture Visits | Culture Employment | Culture Consumption |
|---|---|---|---|---|---|---|
| 2020-06 | 0.147 | 0.259 | 0.193 | 0.425 | 0.274 | 0.207 |
| 2020-07 | 0.109 | 0.157 | 0.143 | 0.191 | 0.181 | 0.121 |
| 2020-08 | 0.132 | 0.153 | 0.168 | 0.164 | 0.147 | 0.134 |
| 2020-11 | 0.098 | 0.144 | 0.185 | 0.265 | 0.125 | 0.131 |
| 2020-12 | 0.097 | 0.116 | 0.242 | 0.192 | 0.099 | 0.098 |
| 2021-09 | 0.133 | 0.133 | 0.178 | 0.134 | 0.166 | 0.165 |
| 2021-10 | 0.218 | 0.265 | 0.218 | 0.277 | 0.338 | 0.281 |
| Average | 0.134 | 0.175 | 0.190 | 0.252 | 0.190 | 0.162 |
| Adding | 0.078 | 0.042 | 0.056 | 0.118 | 0.056 | 0.026 |

Source: Authors' calculation, using data specified in the Data Availability Statement. Note: Regressions with negative values of the coefficients are marked in red. For Culture visits, the average value was calculated without the 2021-09 figure.

## 5. Discussion

While the impact of the COVID-19 on the overall fall in tourism worldwide is evident, it is difficult to model the structural issues laying behind relative outcome by countries. The results of this study were rather surprising showing that incidence of COVID-19 and relative fall in tourism in the selected 20 European countries were not statistically related for 19 out of 20 months under review. A possible explanation is that tourism was not affected only by the relative incidence of the disease by countries, but also by different, country specific prohibitive measures, not captured in this study.

Regarding structural issues determining resilience of tourism in the current pandemic, Pocinho et al. (2022) reviewed 18 recent scientific articles, with a common feature that future of tourism shall be based on positive and resilient approach focused on the tourists' safety and wellbeing, promoting the quality of life. Most likely, the interest of tourists will shift towards the health and wellness tourism (Wen et al. 2020), while they seem to be reluctant to the cruise tourism (Pan et al. 2021). One study (Kock et al. 2020) investigated in more details fear of contracting COVID-19 and found positive relations to indicators of xenophobia, ethnocentrism, crowding perceptions and attitude towards group travel. An estimate (Lee et al. 2022), using data for Taiwan has shown that a 1% increase in the number of confirmed cases of COVID-19 reduced the number of tourist visits by 0.1% while total revenues of the hotels decreased by 0.33%, with rather different impact, depending on the quality of the hotels.

In contrast, in this study, adding the variables of the share of hotels in the total number of overnight stays (depicting a perceived risk due to close encounters with other persons), as well as a share of inbound tourism conducted by land transport (depicting closeness of travel) found very low coefficients of determination in the range 0.10 to 0.15 for only 6 months and moderate (above 0.20) only for a single month out of 20 under review.

However, geographical closeness was stressed as important by the European Travel Commission European Travel Commission Report (2022) stating that Europe benefited from short-haul travel in 2021, led by strong performance of destinations accessible by car from their large source markets, such as Croatia, Denmark, Luxembourg, Slovenia, France and Switzerland. Moreover, Huang et al. (2021), using survey and interviews, found that most tourists have changed their travel preferences towards the countries that are close in geographic as well as cultural sense (in terms of values and attitudes). The same study also

concluded that 82% respondents would prefer to travel to nature-based destinations, 66% to destinations with rich history, culture and cultural heritage and 63% to rural destinations.

This study is in line with that, while adding variables of cultural activities improved the explanatory power of the baseline model in explaining variance of tourism fall by countries. The best fitting variable of culture visits added 11.8 percentage points to the explanatory power of the model, lifting it up to a still moderate 25.2%, on average for the 7 months out of 20 under review. This improved the simple regression model more than the two variables (share of air transport and share of hotels) that seemed to be very important. The other two variables i.e., culture employment and culture consumption added 5.6 and 2.6 percentage points to the explanatory power of the model, respectively.

It is evident from the current study as well as from the reviewed literature that, for explaining relative impact of the pandemic on tourism by countries there is a need for further research into other variables to capture issues such as business travel, family visits as well as the use of facilities owned by tourists, which are presumably less influenced by relative intensity of pandemic. A more thorough research into the phenomenon of substitution of long-haul tourism by close destinations, including those within the countries may be needed as well.

## 6. Conclusions

The research identified possible importance of the relative intensity of cultural activities for the resilience of tourism within the COVID-19 pandemic, adding to the findings of other studies stressing the importance of geographical and cultural closeness between the inbound and outbound countries, as well as possibilities to maintain low health risk due to close contacts with unknown persons.

The analysis presented is subject to the limitations of methods used so the readers are advised to be cautious with interpretation of the results. While this research reveals insightful findings about the possible linkages between culture and impact of the pandemic on tourism for the 20 European countries, the future studies should include larger number of countries as well as additional variables in order to obtain a more thorough insight into the importance of natural and cultural heritage for the resilience of tourism.

**Funding:** This research received no external funding.

**Institutional Review Board Statement:** Not applicable.

**Informed Consent Statement:** Not applicable.

**Data Availability Statement:** Overnight stays at tourist accommodation establishments, 2019–2021: Eurostat monthly data [TOUR_OCC_NIM__custom_2135440] Data extracted on 21 February 2022 from [ESTAT], last updated on 14 February 2022. Monthly data on the daily number of new reported COVID-19 cases, 2020–2021: Data on the daily number of new reported COVID-19 cases and deaths by EU/EEA country, https://www.ecdc.europa.eu/en/publications-data/data-daily-new-cases-covid-19-eueea-country, accessed on 21 February 2022. Inbound tourism: Arrivals by air, percentage of total inbound arrivals, 2019: Tourism Statistics Data, UNWTO https://www.unwto.org/tourism-statistics-data, extracted on 21 February 2022, data for 2019, except for Bulgaria, Estonia, Finland, France, Netherlands, Norway, Portugal and Sweden, for which data were assessed using data from previous years, applying a yearly multiplier (1027) calculated for the countries with full data series. For Austria, Belgium, Czechia, Denmark, Germany, Lichtenstein and Slovakia due to no availability, data were assessed calculating average values of the indicators of two of their respective neighbouring countries. Terrestrial protected area (%), 2020: Eurostat, data extracted on 21 February 2022 from [ESTAT] Dataset: Natura 2000 protected areas (source: EEA) [ENV_BIO1__custom_2137417], last updated on 3 March 2022. Data for Norway Island and Lichtenstein are extracted from national sources. Number of World Heritage sites inscribed by each State Party, 2022: UNESCO World Heritage List Statistics, https://whc.unesco.org/en/list/stat/, accessed on 21 February 2022. Cultural employment by sex, percentage of total employment, 2020: Eurostat, data extracted on 21 February 2022 from [ESTAT], CULT_EMP_SEX__custom_2138832], last updated on 25 June 2021. Mean consumption expenditure of private households on cultural goods and services by COICOP consumption purpose, Percentage of total expenditure, Pur-

chasing power standard (PPS), 2015: Eurostat, data extracted on 21 February 2022 from [ESTAT] CULT_PCS_HBS__custom_2138858], last updated on 6 August 2020. Overnight stays at tourist accommodation establishments, 2019–2021: Eurostat monthly data [TOUR_OCC_NIM__custom_2135440], data extracted on 21 February 2022 from [ESTAT], last updated on 14 February 2022. Monthly data on the daily number of new reported COVID-19 cases, 2020–2021: Data on the daily number of new reported COVID-19 cases and deaths by EU/EEA country, https://www.ecdc.europa.eu/en/publications-data/data-daily-new-cases-covid-19-eueea-country, accessed on 21 February 2022. Inbound tourism; Arrivals by air, percentage of total inbound arrivals, 2019: Tourism Statistics Data, UNWTO https://www.unwto.org/tourism-statistics-data, extracted on 21 February 2022, data for 2019, except for Bulgaria, Estonia, Finland, France, Netherlands, Norway, Portugal and Sweden, for which data were assessed using data from previous years, applying a yearly multiplier (1027) calculated for the countries with full data series. For Austria, Belgium, Czechia, Denmark, Germany, Lichtenstein and Slovakia due to no availability, data were assessed calculating average values of the indicators of two of their respective neighboring countries. Terrestrial protected area (%), 2020: Eurostat, data extracted on 21 February 2022 from [ESTAT] Natura 2000 protected areas (source: EEA) [ENV_BIO1__custom_2137417], last updated on 3 January 2022. Data for Norway Island and Lichtenstein are extracted from national sources. Number of World Heritage sites inscribed by each State Party, 2022: UNESCO World Heritage List Statistics, https://whc.unesco.org/en/list/stat/, accessed on 21 February 2022. Cultural employment by sex, percentage of total employment, 2020: Eurostat, data extracted on 21 February 2022 from [ESTAT], CULT_EMP_SEX__custom_2138832], last updated on 25 June 2021. Mean consumption expenditure of private households on cultural goods and services by COICOP consumption purpose, Percentage of total expenditure, Purchasing power standard (PPS), 2015: Eurostat, data extracted on 21 February 2022 from [ESTAT] CULT_PCS_HBS__custom_2138858], last updated on 6 August 2020. Visits to cultural sites (historical monuments, museums, art galleries or archaeological sites) at least once in the last 12 months, percentage of population 16 years or over, 2015: Eurostat, data extracted on 21 February 2022 from [ESTAT] Frequency of participation in cultural or sport activities in the last 12 months by sex, age, educational attainment level and activity type [ILC_SCP03__custom_2138799], last updated on 20 March 2019.

**Conflicts of Interest:** The author declares no conflict of interest.

## Appendix A

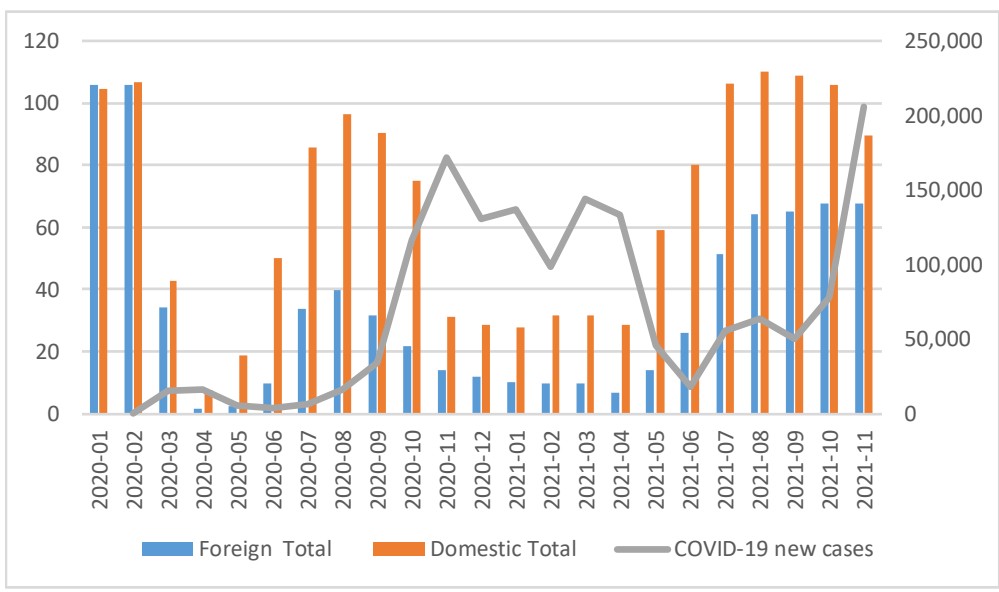

**Figure A1.** Overnight stays at tourist accommodation establishments—European Union 27 countries (EU27) (2019 = 100) and monthly average of new reported COVID-19 cases in EU27 and EFTA countries. Source: Eurostat monthly data [TOUR_OCC_NIM__custom_2135440] extracted on 21 February 2022

from [ESTAT], Last updated 14 February 2022. Data on the daily number of new reported COVID-19 cases and deaths by EU/EEA country, https://www.ecdc.europa.eu/en/publications-data/data-daily-new-cases-covid-19-eueea-country, accessed on 21 February 2022. Note: Norway, Iceland and Liechtenstein were included while Switzerland was not included in the COVID-19 cases numbers.

**Table A1.** Nights spent at tourist accommodation establishments—European Union 27 countries (2019 (1–11) = 100).

| Overnight Stays | 2020 (1–11) | 2021 (1–11) |
| --- | --- | --- |
| **Hotels and similar accommodation** | 44.1 | 54.3 |
| *Foreign* | 27.6 | 36.5 |
| *Domestic* | 61.4 | 73.2 |
| **Holiday and other short-stay accommodation** | 57.3 | 69.2 |
| *Foreign* | 37.6 | 48.0 |
| *Domestic* | 71.7 | 84.8 |
| **Camping grounds, recreational vehicle parks and trailer parks** | 70.3 | 87.6 |
| *Foreign* | 42.4 | 61.2 |
| *Domestic* | 87.9 | 104.2 |
| **Grand Total** | 50.5 | 62.1 |
| *Foreign Total* | 31.2 | 41.5 |
| *Domestic Total* | 68.1 | 80.9 |

Source: Eurostat monthly data [TOUR_OCC_NIM__custom_2135440] Data extracted on 21 February 2022 from [ESTAT], Last updated 14 February 2022.

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
