# Peer review of "Were Culture and Heritage Important for the Resilience of Tourism in the COVID-19 Pandemic?"

_jrfm, doi:10.3390/jrfm15050205_

Round 1

Reviewer 1 Report

Dear Authors,

I find this manuscript interesting as it focuses on identifying possible effects of the Pandemic Covid 19 on tourism outcome by 20 European countries, between 2020 and 2021.

Key words: you should avoid using key words  which are also in the title: tourism, culture, heritage

Introduction:

In introduction it is a repetition: Lines: 22-24: The The fall of tourism due to recent pandemic was overwhelming and unprecedented  in more than 70 years, which especially stands for international arrivals falling 73% in 2020,  and virtually remaining the same in 2021.

No results should be presented in the introduction, no graphics should be inserted. Figure 1 and table 1 should be included to the results section.

I suggest you to divide the Introduction section into:

Introduction (related to the general introduction to the study, the main aim, its implications, and structure of the paper: a brief advance of the parts in which the text is divided)

Literature background: related to the explanation of the impact of Covid 19 Pandemic on the fall of tourism.

Results section

Figure 4 is missing.

You should check the grammar: Lines between 122-126: This The results of simple regression on impact of relative intensity of COVID-19 on  122 tourist overnight stays by the 20 countries under review came rather as surprise. The correlation coefficients for almost the whole 20 months period under review was extremely low, indicating that variance in the relative fall in tourism cannot at all be explained only by different relative incidence of pandemic by countries.

Discussion section

- I think the first word should be removed (Line 219): Authors While the impact of the COVID-19 on the overall fall in tourism worldwide 

-You should better highlight the results obtained. So, the results should be better compared with the results obtained in other studies by reference to similarities or differences. This comparison of the results would better reflect the relevance of the study; you should make a better description of the contribution of the results to the development of the field (theoretical or methodological) emphasizing the importance of the study.

To be added an idea of the future studies and study limitations

Conclusions

The conclusion section is missing.

Author Response

  1. You should avoid using key words  which are also in the title: tourism, culture, heritage
  • Changed as suggested
  1. In introduction it is a repetition: Lines: 22-24: The The fall of tourism due to recent pandemic was overwhelming and unprecedented  in more than 70 years, which especially stands for international arrivals falling 73% in 2020,  and virtually remaining the same in 2021.
  • Corrected
  1. No results should be presented in the introduction, no graphics should be inserted. Figure 1 and table 1 should be included to the results section.
  • Figure 1 and Table 1 moved to Annex
  1. I suggest you to divide the Introduction section into: Introduction (related to the general introduction to the study, the main aim, its implications, and structure of the paper: a brief advance of the parts in which the text is divided) Literature background: related to the explanation of the impact of Covid 19 Pandemic on the fall of tourism.
  • Divided and restructured as suggested
  1. Results section - Figure 4 is missing.
  • Inserted back. It was lost in transfer to the template.
  1. You should check the grammar: Lines between 122-126: This The results of simple regression on impact of relative intensity of COVID-19 on  122 tourist overnight stays by the 20 countries under review came rather as surprise. The correlation coefficients for almost the whole 20 months period under review was extremely low, indicating that variance in the relative fall in tourism cannot at all be explained only by different relative incidence of pandemic by countries.
  • Corrected
  1. The first word should be removed (Line 219): AuthorsWhile the impact of the COVID-19 on the overall fall in tourism worldwide 
  • Corrected
  1. You should better highlight the results obtained. So, the results should be better compared with the results obtained in other studies by reference to similarities or differences. This comparison of the results would better reflect the relevance of the study; you should make a better description of the contribution of the results to the development of the field (theoretical or methodological) emphasizing the importance of the study.
  • Done as suggested
  1. To be added an idea of the future studies and study limitations
  • Added as suggested
  1. The conclusion section is missing.
  • The conclusion section added

Reviewer 2 Report

-86 ord- "Some studies..." cannot be said and then only one source cited
-Sources must be cited in the first three paragraphs
-There are technical and spelling errors
-Excess words at the beginning of chapters 1, 3 and 4
-It is necessary to insert a chapter Literature review
-There are only 8 sources in the paper, and 20 sources at the end of the paper. All listed sources at the end of the paper must be also in the paper
-Expand sources
-Technically arrange / unify the citation of literature

Author Response

86 ord- "Some studies..." cannot be said and then only one source cited

  • Corrected

 Sources must be cited in the first three paragraph

  • Corrected

 There are technical and spelling errors

  • Text checked and corrected by expert

 Excess words at the beginning of chapters 1, 3 and 4

  • Corrected

It is necessary to insert a chapter Literature review

  • Inserted as suggested

There are only 8 sources in the paper, and 20 sources at the end of the paper. All listed sources at the end of the paper must be also in the paper. Expand sources

  • Now the number of sources is the same within the paper and at the end

Technically arrange / unify the citation of literature

  • Done

Round 2

Reviewer 1 Report

I appreciate the effort made by the authors for improving the manuscript. One detail I would like to point out in the section of literature review: the word chapter to be replaced by section:

Lines102-103: "The literature dealing with possible impacts of natural and cultural heritage on the resilience to crises is very limited and will be pointed out to in the discussion chapter".

Author Response

Thank you very much for your help in revision of the text.

I have changed the word "chapter" into "section" as suggested.

Reviewer 2 Report

-The author has made all the suggested corrections and improved the manuscript

Author Response

Thank you very much for kind and valuable assistance in reviewing the document.